# SwitchHead: Accelerating Transformers with Mixture-of-Experts Attention

**Róbert Csordás**[1†]   **Piotr Piękos**[2]   **Kazuki Irie**[3†]   **Jürgen Schmidhuber**[2,4]

[1]Stanford University, Stanford, CA, USA
[2]AI Initiative, KAUST, Thuwal, Saudi Arabia
[3]Center for Brain Science, Harvard University, Cambridge, MA, USA
[4]The Swiss AI Lab IDSIA, USI & SUPSI, Lugano, Switzerland
rcsordas@stanford.edu, piotr.piekos@kaust.edu.sa,
kirie@fas.harvard.edu, juergen@idsia.ch

## Abstract

Despite many recent works on Mixture of Experts (MoEs) for resource-efficient Transformer language models, existing methods mostly focus on MoEs for *feedforward* layers. Previous attempts at extending MoE to the *self-attention* layer fail to match the performance of the *parameter-matched* baseline. Our novel SwitchHead is an effective MoE method for the *attention layer* that successfully reduces both the compute and memory requirements, achieving wall-clock speedup, while matching the language modeling performance of the baseline Transformer. Our novel MoE mechanism allows SwitchHead to compute up to 8 times fewer attention matrices than the standard Transformer. SwitchHead can also be combined with MoE feedforward layers, resulting in fully-MoE "SwitchAll" Transformers. For our 262M parameter model trained on C4, SwitchHead matches the perplexity of standard models with only 44% compute and 27% memory usage. Zero-shot experiments on downstream tasks confirm the performance of SwitchHead, e.g., achieving more than 3.5% absolute improvements on BliMP compared to the baseline with an equal compute resource.[1]

## 1   Introduction

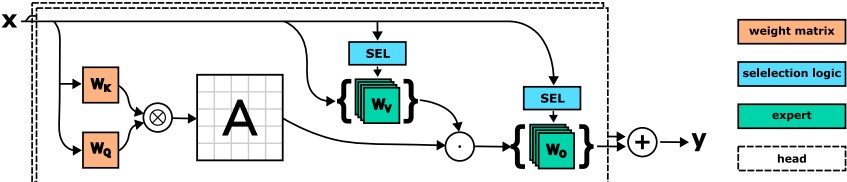

Figure 1: A schematic representation of SwitchHead. It consists of a few independent heads, each with multiple experts for value and output projections. Each head has a single attention matrix.

Large language models (LLMs) have shown remarkable capabilities [1, 2, 3, 4] and great versatility [5]. However, training large Transformers [6, 7] requires a considerable amount of computing power and memory, which is not accessible to most researchers, academic institutions, and even companies.

---

[†]Work done at IDSIA.
[1]Our code is public: https://github.com/robertcsordas/switchhead

38th Conference on Neural Information Processing Systems (NeurIPS 2024).

Even running them in inference mode—typically much less resource-intensive—requires significant engineering effort [8]. Accelerating Transformers remains an important research question.

In this context, Mixture of Experts (MoE) layers [9, 10, 11] have become popular to efficiently scale up Transformers to a large number of parameters [12, 13, 14, 15, 16, 17]. However, most of these works mainly focus on applying MoE to the 2-layer *feedforward blocks* [6], i.e., the multi-layer perceptron (MLP) components of the Transformer, while keeping the self-attention layers unchanged. Given that attention also accounts for a considerable amount of compute and memory usage in Transformers (especially for long context sizes), using *MoE for attention* has potential to further improve resource efficiency in Transformers. While MoE-based attention remains underexplored in general, there are existing works on MoE approaches for attention [18, 19]. However, in practice, previously proposed methods typically require a lot of engineering tricks for successful training, *and* most importantly, only achieve a modest reduction in computing and memory requirements in the end (as we also confirm in our experiments).

Here, we present a novel MoE-based attention method, SwitchHead, whose mechanism allows to reduce the number of attention matrices that need to be computed and stored. Following $\sigma$-MoE [17], our method uses a non-competitive selection activation function (sigmoid), and does not require regularization or extra tricks for stable training. Importantly, we show that it is possible to compute the MoE projections *outside* of the attention core, which enables a significant reduction in the number of computed attention maps, resulting in significant resource savings. Our thorough investigation shows that it is enough to choose the value and output projections from a pool of experts and share keys and queries between them.

We evaluate our method on C4 [20], Enwik8 [21], peS2o [22] and Wikitext 103 [23], with two model sizes (47M and 262M). Additionally, we measure the zero-shot performance of our main models on Lambada [24], BLiMP [25], and Children's Books Test [26] datasets. Our experiments demonstrate that SwitchHead can achieve performance comparable to parameter-matched baselines with just a fraction of the compute and memory budget. In addition, we introduce "SwitchAll", a fully MoE-based Transformer model, that combines a $\sigma$-MoE-based MLP layer with our SwitchHead attention, often outperforming dense baselines with the same parameter budgets.

Finally, we analyze the attention maps of our SwitchHead. We find that the attention maps taken over all heads are qualitatively similar to the dense baselines, indicating a significant reduction in redundancy without a loss of expressivity. In addition, expert selections are often interpretable.

## 2 Method

### 2.1 Background

The standard multi-head self-attention (MHA) layer [6] consists of four major steps: (1) compute key, query, and value projections, (2) compute the attention matrix, (3) use the attention matrix to project the values, and (4) map the projected values to the output. Let $h$, $T$, $n_{\text{heads}}$, $d_{\text{model}}$, $d_{\text{head}}$ denote positive integers. Let $x \in \mathbb{R}^{T \times d_{\text{model}}}$ denote an input to the MHA layer with $n_{\text{heads}}$ heads, $T$ be the sequence length, and $d_{\text{model}}$ denote the size of the hidden representations of the model. $W_{\{K,V,Q\}}^h \in \mathbb{R}^{d_{\text{model}} \times d_{\text{head}}}$ are the projection matrices for head $h \in \{1, ..., n_{\text{heads}}\}$. Then $K^h = xW_K^h$, $Q^h = xW_Q^h$, and $V^h = xW_V^h$ (thus $K^h, Q^h, V^h \in \mathbb{R}^{T \times d_{\text{head}}}$) are the keys, queries, and values, respectively. The attention matrix for the head $h$, $A^h \in \mathbb{R}^{T \times T}$, and the output $y \in \mathbb{R}^{T \times d_{\text{model}}}$ are calculated as follows:

$$A^h = \text{softmax}\left(\frac{1}{\sqrt{d_{\text{head}}}} Q^h K^{h\mathsf{T}}\right) \tag{1}$$

$$y = (A^1 V^1 | A^2 V^2 | ... | A^{n_{\text{heads}}} V^{n_{\text{heads}}}) W_O \tag{2}$$

where $|$ denotes concatenation in the last dimension, the $\text{softmax}(\cdot)$ is also over the last dimension, and $W_O \in \mathbb{R}^{n_{\text{heads}} d_{\text{head}} \times d_{\text{model}}}$. However, an alternative formulation reflects the role of $W_O$ better. Let us divide $W_O$ along the first dimension into submatrices for each head, $W_O^h \in \mathbb{R}^{d_{\text{head}} \times d_{\text{model}}}$, such that $W_O = \left(W_O^{1\mathsf{T}} | W_O^{2\mathsf{T}} | ... | W_O^{n_{\text{heads}}\mathsf{T}}\right)^{\mathsf{T}}$. In this case, the output (Eq. 2) can be equivalently written as:

$$y = \sum_h A^h V^h W_O^h \tag{3}$$

From this, it can be seen that all computations are local to each head. Computing the attention matrix $\boldsymbol{A}^h$ and the readout $\boldsymbol{A}^h\boldsymbol{V}^h$ requires compute in order of $O(n_{\text{heads}}d_{\text{head}}T^2)$ MACs (multiplication-accumulation operation[2]). During training, it requires the storage of $O(n_{\text{heads}}T^2)$ for the attention matrices and $O(n_{\text{heads}}Td_{\text{head}})$ for storing the sub-results of the projections. Given a sufficiently long sequence, computing the attention matrix and projecting the values will dominate the compute requirements due to the quadratic dependence on the sequence length $T$.

## 2.2    From Dense to SwitchHead Attention Layer

Our goal is to obtain resource reductions while maintaining the fundamental properties of attention and retaining a fully expressive attention matrix. For that, we start from the following observation: modern LLMs use tens of heads [2, 27]. Are so many of them all necessary? As we show later in Sec. 3, indeed, naively reducing the number of heads (while keeping the same number of parameters by increasing the head dimension) results in performance loss. Explaining the reason for the need for many heads is beyond the scope of this paper. Nevertheless, here are some hypotheses: (1) they provide multiple inputs for the operations that the network performs in each step, (2) they are specialized and provide inputs only for specific operations (in this case, each operation would use a different subset of heads), (3) they may provide diverse outputs due to different initializations, some being more successful than others, thus enabling better learning. Among these, (2) and (3) may offer an opportunity for resource savings: if not all heads are needed at the same time, it might be possible to *switch* among them depending on the context.

One naive method to achieve this is to use a gating signal using a linear projection $\boldsymbol{W}_S \in \mathbb{R}^{d_{\text{model}} \times n_{\text{heads}}}$, and use the heads with the highest score, by replacing Eq. 3 with Eq. 6:

$$\boldsymbol{s} = \sigma\left(\boldsymbol{x}\boldsymbol{W}_S\right) \tag{4}$$

$$\mathcal{E} = \arg\text{topk}(\boldsymbol{s}, k), \mathcal{E} \subset \{1, ..., n_{\text{heads}}\} \tag{5}$$

$$\boldsymbol{y}[t, c] = \sum_{h \in \mathcal{E}} \boldsymbol{s}[t, h](\boldsymbol{A}^h\boldsymbol{V}^h\boldsymbol{W}_O^h)[t, c] \tag{6}$$

where $\boldsymbol{y}[t, c] \in \mathbb{R}$ denotes indexing the specific element of the output matrix $\boldsymbol{y} \in \mathbb{R}^{T \times d_{\text{model}}}$, for timestep $t$ and channel $c$, and $k$ is the number of active experts. Following the $\sigma$-MoE method [17], we use a non-competitive selection function (sigmoid $\sigma$ in Eq. 4). Now, let us define the *source* side of attention as the keys and values and the *destination* side as the queries and output. Intuitively, the above method corresponds to choosing a subset of attention heads based on the *destination* side alone[3]. Our preliminary experiments confirmed that this method is indeed feasible for language modeling on WikiText-103. However, it is difficult to achieve acceleration and memory savings with this method. To see why, notice that the entries of the attention matrix $\boldsymbol{A}^h$ depend on *pairs* of tokens, one for the source and one for the destination side, but the choice is made *only* based on the destination side. Thus, in the worst case, for each destination, a different source might be chosen, in which case all possible source projections have to be computed for the keys and values, which we would like to avoid.

Alternatively, we propose to improve the method above by introducing conditional computations for the source and destination projections independently of each other. That is, we parameterize each of key, query, value, output projection by an independent MoE. This avoids conditional computations that involve the attention matrix itself. Our solution implements this using Mixtures of Experts (MoEs). The concepts of "heads" are no longer well defined in the conventional sense: we redefine a head as an instance of a computed attention matrix. We call the total number of them $n_{\text{heads}}$. For each head $h$, we define a separate list of $E$ experts. The total number of *experts* is then $n_{\text{heads}} \cdot E$. Then, the projection matrices become $\boldsymbol{W}_K^{h,e}$, $\boldsymbol{W}_Q^{h,e}$, $\boldsymbol{W}_V^{h,e}$ and $\boldsymbol{W}_O^{h,e} \in \mathbb{R}^{d_{\text{head}} \times d_{\text{model}}}$, where $h$ denotes the head index and $e$ the specific expert. Then we compute the source-side expert selection as follows:

$$\boldsymbol{s}_S^h = \sigma(\boldsymbol{x}\boldsymbol{W}_S^h) \tag{7}$$

$$\mathcal{E}_S^h = \arg\text{topk}(\boldsymbol{s}_S^h, k), \mathcal{E}_S^h \subset \{1, ..., E\} \tag{8}$$

---

[2]The number of MACs is a metric used in prior work [18], which is independent of both the specific hardware and implementation, unlike wall-clock time. For wall-clock-time measurements, see Sec. 3.7.

[3]To clarify, we allocate a routing function for each of key/value/query projections; these routing functions belong to the *source* or *destination* side accordingly. If we compare Eq. 10 and Eq. 6, one can notice that the routing function in Eq. 6 effectively corresponds to what we define as the *destination*-side routing in Eq. 10.

where $\boldsymbol{W}_S^h \in \mathbb{R}^{d_{\text{model}} \times E}$. We compute the destination-side experts similarly: $\boldsymbol{s}_D^h = \sigma(\boldsymbol{x}\boldsymbol{W}_D^h)$, $\mathcal{E}_D^h = \arg\text{topk}(\boldsymbol{s}_D^h, k), \mathcal{E}_S^h \subset \{1, ..., E\}, \boldsymbol{W}_D^h \in \mathbb{R}^{d_{\text{model}} \times E}$. Then, the value projection $\boldsymbol{V}^h$ is computed as a weighted sum of the selected experts:

$$\boldsymbol{V}^h = \sum_{e \in \mathcal{E}_S^h} \boldsymbol{s}_S^h[e]\boldsymbol{x}\boldsymbol{W}_V^{h,e} \tag{9}$$

The key and query projections are computed similarly: $\boldsymbol{K}^h = \sum_{e \in \mathcal{E}_S^h} \boldsymbol{s}_S^h[e]\boldsymbol{x}\boldsymbol{W}_K^{h,e}$, and $\boldsymbol{Q}^h = \sum_{e \in \mathcal{E}_D^h} \boldsymbol{s}_D^h[e]\boldsymbol{x}\boldsymbol{W}_Q^{h,e}$. The output projection also becomes an MoE:

$$\boldsymbol{y} = \sum_{h=0}^{n_{\text{heads}}-1} \sum_{e \in \mathcal{E}_D^h} \boldsymbol{s}_D^h[e]\boldsymbol{A}^h\boldsymbol{V}^h\boldsymbol{W}_O^{h,e} \tag{10}$$

As we'll show, it is not necessary to make all projections MoEs. In Section 3.1 we show that keeping a single, head-specific copy of the query and key projections and reusing them for all experts is beneficial. We call this method SwitchHead.

Essentially, SwitchHead reduces the number of attention matrices that have to be computed ($n_{\text{heads}}$) significantly, by using multiple experts per head. Note that our method does not depend on the specific implementation of the attention, allowing for easy experimentation and research. A schematic representation is shown in Figure 1.

Table 1: Performance of SwitchHead compared to different MoA variants. MoA can outperform the baseline, but only at a price of using significantly more compute and memory. Also, SwitchHead outperforms the baseline dense Transformer. These results are on Wikitext 103. Table sorted by model perplexity.

| #total params | Model | $n_{\text{heads}}$ | Perplexity ↓ | MACs | Mem (floats) |
|---|---|---|---|---|---|
| 47M | SwitchHead | 2 | 12.27 | 170.4M | 0.8M |
| | Transformer | 10 | 12.31 | 453.4M | 3.5M |
| | MoA | 4 | 12.60 | 223.5M | 1.3M |
| | MoA | 6 | 12.64 | 306.8M | 1.9M |
| | MoA | 8 | 12.77 | 390.2M | 2.6M |
| | MoA | 2 | 12.84 | 140.1M | 0.7M |
| 262M | MoA | 8 | 9.50 | 2.9G | 9.9M |
| | SwitchHead | 2 | 9.55 | 2.0G | 2.9M |
| | Transformer | 16 | 9.66 | 5.4G | 21.0M |
| | MoA | 12 | 9.68 | 4.1G | 14.7M |
| | MoA | 4 | 9.69 | 1.7G | 5.1M |
| | MoA | 2 | 9.87 | 1.1G | 2.7M |

## 3 Experiments

We conduct our experiments in a *parameter-matched* setting [17] which better reflects the task of language modeling (than the FLOPS-matched setting often used to evaluate MoEs). Our main experiments use Transformer XL, because we found them to consistently and significantly outperform RoPE-based baselines [28] for a fixed amount of compute. We provide the details of this analysis in Appendix A.4. The conclusions on the effectiveness of SwitchHead are consistent in both cases.

As an important specification, under this parameter-matched setting, we always configure Switchhead such that it *matches the perplexity* of the baseline dense Transformer, and we *maximize its resource reductions*. For this, we follow a systematic procedure. First, we set $n_{\text{heads}} * E$ to be the same as $n_{\text{heads}}$ of the dense baseline. We start with setting $n_{\text{heads}} = 2$ and $k = 2$, which provide the most resource reductions. If the resulting model underperforms, we increase $k$. If $k = 4$ underperforms as well, we set $n_{\text{heads}} = 4$ and $k = 2$. We always set $d_{\text{head}}$ so that the total number of parameters of the resulting model matches the number of parameters of the baseline. This reasonably simple procedure ensures a good amount of resource savings, while avoiding doing an expensive hyperparameter search.

Note that all the perplexity gains seen in the main result tables are the byproduct of imperfect matching, and our goal is to achieve *reductions in resource requirements*, unless noted otherwise (See Sec. 3.5). Detailed hyperparameters of all our models can be found in Sec. A.5 in the Appendix. We use and adopt the Triton kernel of $\sigma$-MoE [17] for our purposes.

For all datasets except the character-level Enwik8 [21], we use sub-word units [29, 30] obtained with a SentencePiece tokenizer [31] with a vocabulary size of 8k tokens. For most of our experiments, we use Transformer XL [32] with the context size being twice the size of the active/current chunk, because we found it to be significantly more resource-efficient than the standard setup. However, in order to show that our method is also competitive in the standard Transformer with RoPE positional ecodings, we also demonstrate our main findings in this setup (Appendix A.4).

All models are trained for 100k batches. Some of the datasets we consider (C4 [20], and peS2o [22]) are much larger. In this case, we train on the first $10^5 * T * N_{\text{batch}}$ tokens of the dataset.

## 3.1 Which Projections Require an MoE?

As discussed in Sec. 2.2, each linear projection (keys, values, queries, and output) can potentially be replaced independently by an MoE. Here we first check which projection benefits from such a replacement. As we target the parameter-matched setting, using MoE where it is not necessary can have a negative effect. Since experts use a significant part of the parameter budget, they can reduce the number of parameters available for the more useful parts of the model. Thus, we did a search over all possible combinations of MoE versus fixed projections with two active heads and compared them to the parameter-matched baseline. We find that the output projection is necessary to match the performance of the baseline (for detailed results refer to Tab. 6 in the appendix). Having MoE in the key and query projections turn out to be *un*necessary. Models without the output and value MoE underperform the dense baseline with $n_{\text{heads}} = 2$ heads.

In sum, the best-performing model is the one using MoE for value and output projections. We use this model variant in the rest of experiments in this paper.

## 3.2 Comparison with MoA

The method most related to ours is the so-called Mixture of Attention Heads, or MoA [18]. Unlike SwitchHead, MoA uses a *single* key and value projection and chooses $n_{\text{heads}}$ active query and output projections from a pool of $E$ experts.

MoA computes the attention map for each selected expert and computes their weighted average after the attention computation takes place. In contrast, SwitchHead calculates the weighted average of the $K$ selected experts *before* and *after* attention computation. Because of this, in practice, the same perplexity is achieved with the required number of computed attention matrices ($n_{\text{heads}}$) which is much lower for SwitchHead compared to MoA, allowing significant resource savings.

Also, unlike MoA, SwitchHead uses a non-competitive activation function (sigmoid) [17]. We confirm that with this, our method performs well without any regularization, while MoA requires three different regularizers.

We compare our method with MoA in Table 1. It can be seen that while MoA can slightly outperform our method in terms of perplexity, it can only do so at the price of significantly more resource usage. Given a similar computation and memory budget, our method consistently outperforms MoA.

## 3.3 Performance on Different Datasets

We test our methods on a diverse set of language modeling datasets, including C4 [20], Enwik8 [21], peS2o [22], at two different scales: a 47M and a 262M parameters. We chose this experimental setting taking into account our compute-budget and confidence in our results which are consistent in across various configurations.

The results are shown in Table 2. We compare our models to two baselines: one with the same number of heads as the total number of experts ($n_{\text{heads}} \cdot E$) of the SwitchHead models, and the other has the same number of heads as the number of active attention matrices ($n_{\text{heads}}$) as our models. Our

Table 2: Performance of SwitchHead compared to baselines on different datasets and model sizes. It can be seen that the predictive performance of our SwitchHead model is comparable to the baselines, and is always better than the baseline with an equal number of heads. Perplexity is shown for Wikitext 103, C4 and peS2o datasets, and bits/character (bpc) for Enwik8. Models sorted by perplexity.

| Dataset | #total params | Model | $n_{\text{heads}}$ | ppl/bpc ↓ | MACs | Mem (floats) |
|---|---|---|---|---|---|---|
| C4 | 47M | SwitchHead | 2 | 22.53 | 203M | 0.8M |
| | | Transformer | 10 | 22.71 | 453M | 3.5M |
| | | Transformer | 2 | 23.71 | 453M | 1.4M |
| | 262M | SwitchHead | 4 | 16.23 | 2.4G | 5.6M |
| | | Transformer | 16 | 16.28 | 5.4G | 21M |
| | | Transformer | 4 | 17.09 | 5.4G | 8.4M |
| Wikitext 103 | 47M | SwitchHead | 2 | 12.31 | 170M | 0.8M |
| | | Transformer | 10 | 12.32 | 453M | 3.5M |
| | | Transformer | 2 | 12.73 | 453M | 1.4M |
| | 262M | SwitchHead | 2 | 9.77 | 2.0G | 2.9M |
| | | Transformer | 16 | 9.80 | 5.4G | 21M |
| | | Transformer | 2 | 10.09 | 5.4G | 6.3M |
| peS2o | 47M | Transformer | 10 | 12.83 | 453M | 3.5M |
| | | SwitchHead | 2 | 12.84 | 203M | 0.8M |
| | | Transformer | 2 | 13.37 | 453M | 1.4M |
| | 262M | Transformer | 16 | 9.78 | 5.4G | 21M |
| | | SwitchHead | 4 | 9.86 | 2.4G | 5.6M |
| | | Transformer | 4 | 10.11 | 5.4G | 8.4M |
| Enwik8 | 41M | Transformer | 8 | 1.10 | 1.6G | 10M |
| | | SwitchHead | 2 | 1.10 | 709M | 2.8M |
| | | Transformer | 2 | 1.13 | 1.6G | 4.2M |

models closely match the performance of the full, many-head baseline with the fraction of memory and compute requirements (see Sec. 3.7 for more details).

In addition, we verify the performance of our models trained on the C4 dataset downstream tasks in a zero-shot manner. We consider Lambada [24], BLiMP [25] and Children's Book Test (CBT) [26]. The results are shown in Table 4: our SwitchHead models consistently outperform or match the performance of the baseline dense Transformer models.

## 3.4 SwitchAll

The goal of achieving more resource-efficient Transformers includes reducing the resource requirements of both the MLP and the attention layers. $\sigma$-MoE [17] was recently proposed as a parameter-efficient MoE method for accelerating the MLP layers. However, it remains unclear whether it can be efficiently combined with our SwitchHead, or can have some negative interaction effect if combined in a "SwitchAll", where every layer is MoE-based.

To verify this, we take the baseline architecture of Csordás et al. [17] without any hyperparameter change and replace the attention layer with SwitchHead. The hyperparameters for the attention are directly taken from the experiments shown in Tab. 2. The results are shown in Tab. 3. The combined, fully-MoE model often outperforms the dense baselines for each dataset and model size considered, except in the case of the 262M parameter model on the C4 dataset.

## 3.5 MAC-Matched Setup

All our experiments so far were calibrated so that the predictive performance (perplexity) matches to the performance of the baseline Transformer, and we were aiming for maximum resource savings. However, it is also a valid question to ask what is the performance of SwitchHead in a MAC-matched setup, where the compute requirements of our model are matched to those of the baseline. We achieve this by increasing $d_{\text{head}}$ and $n_{\text{heads}}$ until we have the same MAC requirements as the baseline. This results in a model with more parameters. For the small Transformer XL, we increase $d_{\text{head}}$ from 76 to

112 and $n_{\text{heads}}$ from 2 to 3. For large XL, we increase $n_{\text{heads}}$ from 4 to 6 and $d_{\text{head}}$ from 112 to 168. For the small RoPE model, we change $n_{\text{heads}}$ from 2 to 3 and $d_{\text{model}}$ from 64 to 84, for big $n_{\text{heads}}$ from 4 to 6 and $d_{\text{model}}$ from 112 to 168. We show the results in Tab. 4: MAC-matched models outperform the others by a large margin both in perplexity and in zero-shot task performance.

### 3.6 Shared Selection

For further time savings, we can share the expert selection between the source and destination side. Acceleration is achieved by reducing the number of sorting and top-k steps compared to the full SwitchHead. However, this results in a minor performance loss, which might be tolerated in some cases where the acceleration is more important. See Tab. 4 for more details.

### 3.7 Wall-Clock Time and Memory Usage Estimation

In all of our tables, we report the number of multiply-accumulate (MAC) operations following Zhang et al. [18]. The reason for this is that the actual wall-clock time is highly implementation and hardware-dependent. Nevertheless, we measured the runtime and total memory usage of our entire training pipeline (including the feedforward layer) to demonstrate that our current (suboptimal) implementation is already capable of providing wall-clock time acceleration. We show the results in Tab. 5. The measurements are taken on identical hardware with the same implementation (including for the attention core), the only difference being the MoE-based projections for the attention. It can be seen that for both scales, SwitchHead trains around 1.5 times faster, while using 61%-67% as much memory as the baseline.

We also report the performance of MoA for reference in Table 5. For measuring the resource usage of MoA, we chose the fastest MoA model that can match the performance of the dense baseline, or simply the best MoA model when no MoA model can match the baseline performance. This resulted in choosing MoA with $H = 4$ for the 47M model and MoA with $H = 8$ for the 262M parameter model. SwitchHead outperforms MoA on both scales, both in wall clock time and memory requirements. Note that these measurements also include the MLP layers, the optimizer, and the gradient synchronization in the case of multi-GPU training.

Table 3: Performance of SwitchAll (SwitchHead + $\sigma$-MoE [17]) on different datasets and model sizes. Our SwitchAll model is close or better compared to the baselines. Models sorted by perplexity. Note: We show the parameter count of the dense model. The parameter count for the big SwitchAll model is 259M because of the imperfect parameter matching.

| Dataset | #total params | Model | $n_{\text{heads}}$ | ppl $\downarrow$ | MACs | Mem (floats) |
|---|---|---|---|---|---|---|
| Wikitext 103 | 47M | SwitchAll | 2 | 12.17 | 170M | 0.8M |
| | | Transformer | 10 | 12.32 | 453M | 3.5M |
| | 262M | Transformer | 16 | 9.80 | 5.4G | 21M |
| | | SwitchAll | 4 | 9.81 | 2.4G | 5.6M |
| C4 | 47M | SwitchAll | 2 | 22.09 | 202M | 0.8M |
| | | Transformer | 10 | 22.63 | 453M | 3.5M |
| | 262M | SwitchAll | 4 | 16.45 | 2.4G | 5.6M |
| | | Transformer | 16 | 16.58 | 5.4G | 21M |
| peS2o | 47M | SwitchAll | 2 | 12.56 | 202M | 0.8M |
| | | Transformer | 10 | 12.83 | 453M | 3.5M |
| | 262M | Transformer | 16 | 9.78 | 5.4G | 21M |
| | | SwitchAll | 4 | 9.86 | 2.4G | 5.6M |

## 4 Analysis

In order to see how the network uses the attention heads, we trained a small, 6-layer, 8-head Transformer on ListOps [33, 34]. The reason for this choice is that small, algorithmic tasks tend to be more interpretable compared to language modeling tasks. We also train a parameter-matched, 2-head

Table 4: Performance of SwitchHead trained on C4 dataset, compared to dense Transformer baseline with matched number of parameters.

| Model | #total params | ppl ↓ | Lambada ↑ | BLiMP ↑ | CBT ↑ |
|---|---|---|---|---|---|
| SwitchHead | 47M | 22.53 | 20.4% | 75.7% | - |
| Transformer | 47M | 22.71 | 20.4% | 73.6% | - |
| SwitchHead MAC-matched | 63M | 21.18 | 23.5% | 77.1% | - |
| SwitchHead Shared selection | 47M | 22.81 | 20.0% | 74.6% | - |
| SwitchHead | 262M | 16.23 | 29.4% | 79.6% | 83.3% |
| Transformer | 262M | 16.28 | 28.2% | 76.1% | 83.6% |
| SwitchHead MAC-matched | 376M | 15.43 | 30.2% | 79.4% | 84.2% |
| SwitchHead Shared selection | 262M | 16.49 | 28.6% | 79.4% | 82.7% |

Table 5: Real-world resource usage of our method. The numbers shown below are for training time for the whole pipeline, including the feedforward layers. It can be seen that SwitchHead in the current implementation reduces both the runtime and the memory usage by a factor of 1.4-1.5.

| Size | Model | ms/iteration | Rel. iter. time | RAM/GPU | Rel. Mem. | #GPUs | GPU type |
|---|---|---|---|---|---|---|---|
| 47M | Transformer | 473ms/iter | 1.0 | 20.5G | 1.0 | | |
| | SwitchHead | 342ms/iter | **0.72** | 13.5G | **0.65** | 1 | RTX 3090 |
| | MoA | 412ms/iter | 0.87 | 15.3G | 0.75 | | |
| 262M | Transformer | 670ms/iter | 1.0 | 20.5G | 1.0 | | |
| | SwitchHead | 442ms/iter | **0.65** | 12.5G | **0.61** | 8 | V100 |
| | MoA | 851ms/iter | 1.27 | 16.4G | 0.80 | | |

SwitchHead model. Both models achieve around 95% accuracy on a held-out IID validation set, in contrast to the dense 2-head model, which saturates around 80%. Note that ListOps is a classification task and does not use autoregressive masking.

We visualize the maximum of attention heads for each layer, both for the standard Transformer (Fig. 2a) and SwitchHead (Fig. 2b). The attention maps are qualitatively similar. Due to different initialization and learning dynamics, thus the overlap between the two models would not be perfect. Complete attention map visualizations can be found in Fig. 4 and 3 in the appendix.

In addition, we anlyze individual attention heads for SwitchHead. We find that it is often possible to interpret the selection weights: on synthetic tasks, the output experts specialize according to different *operations*, while the input ones distinguish numbers and closed parentheses. The attention map itself appears to distribute information about contiguous chunks of numbers (see Fig. 5 in the appendix).

Attention maps of the language models are more difficult to interpret. However, we visualize the attention maps of the 47M parameter Transformer XL and the SwitchHead model from Tab. 2. We find them to be qualitatively similar. We also identified induction heads [35] in both models, some examples shown for SwitchHead in Fig. 6a and for Transformer in Fig. 6b in the appendix. Other typical vertical line-lined attention patterns are shown in Fig. 6c and 6d.

## 5   Related Work

The method most closely related to ours is MoA [18], which introduces a MoE style attention. It defines each attention head as an expert but shares the key and value projections between them. Unlike in our Switchhead, each of the selected experts requires a separate attention matrix, which significantly increases its memory usage. Due to the use of a competitive softmax-based activation function in the selection network, it requires complex regularization to prevent expert collapse [17]. In the original formulation, the number of active heads is high. Our experiments also confirm that MoA needs many attention heads to match the performance of the dense baseline (see Sec. 3.2), and it is only possible to do so with a significantly higher resource budget than our method.

Nguyen et al. [36] analyze the attention matrices, and they conclude that they are usually low rank. Motivated by this, the authors construct a few (e.g., 2) "global attention matrices", and they compute each local matrix for specific heads by a weighted average of those. However, they average the logits,

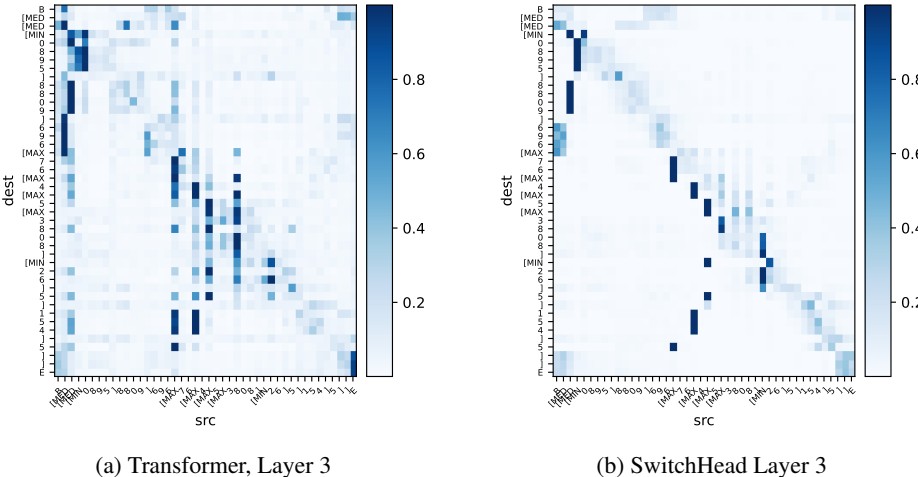

(a) Transformer, Layer 3          (b) SwitchHead Layer 3

Figure 2: An attention map of the (a) standard Transformer and (b) SwitchHead. The maximum of all heads in the given layer are shown.

not the final matrix, so each individual head-specific matrix has to be computed. This means that in the best case, they can only save half of the computation associated with the attention matrix because the readout (Eq. 3) is still needed. For the same reason, memory savings are also low.

Peng et al. [19] propose to reweight the contribution of each head by a gating function. However, they only reduce the number of total attention heads by one, presumably to compensate for the parameters used by the selection logic. Their goal was not to reduce resource usage but to have better predictive performance, which they achieve. They use a softmax-based competitive selection mechanism. To avoid collapse, the gating function is trained only in some steps.

More broadly, there have been several works on MoE to accelerate language models. Shazeer et al. [11] introduce sparsely-gated mixture of experts. Fedus et al. [37] introduce Mixture of Experts in Transformers. Lepikhin et al. [13] train a MoE-based LLM, and Clark et al. [15] analyze the scaling laws of MoE models. Lewis et al. [12] introduce an alternative method for preventing collapse. However, none of these methods focus on the important, *parameter-matched* setting. Csordás et al. [17] introduce the non-competitive activation based MoE method, $\sigma$-MoE, which was shown to be successful in such a setting, but the authors only focused on accelerating the MLPs and not the attention.

Multi-Query attention [38] uses a single key and value projection that is shared between the heads while using multiple queries. Our findings show that such a configuration is suboptimal: using multiple output and value projections is the most important choice in our model design.

Dao et al. [39] provides a hardware-aware CUDA implementation of the entire attention layer, which avoids storing the attention matrix. By saving memory bandwidth in this way, they achieve a significant wall clock time speedup, despite that the attention matrix should be recomputed in the backward pass. This is orthogonal to our method and they can be combined for further acceleration.

## 6  Limitations

Our models are modest in size compared to the current state-of-art LLMs. However, training such models is estimated to cost millions of dollars, which we cannot afford. Instead, we aim to show the versatility of our model by choosing a diverse set of datasets, including Enwik 8, Wikitext 103, C4 and peS2o, and different positional encodings, such as Transformer-XL-style relative positional encoding and RoPE. We also demonstrate the competitiveness of our models in zero-shot downstream tasks. We believe that the evidence we provided is enough for a research group with a larger amount of resources at their disposal to verify our findings in a state-of-the-art model.

The Triton kernel that we used is currently around 60% of the speed of a single dense matrix multiplication of the size of a single expert with cuBLAS. Even this, we showed wall-clock time speedup. We estimate that 80-90% should be achievable with a more optimal kernel. Model-parallel training requires the implementation of a load-balancing system that can dynamically move experts between GPUs.

## 7   Conclusion

On a wide range of language modeling datasets with different model sizes, our novel Mixture-of-Experts (MoE) based attention method called SwitchHead achieves performance of parameter-matched dense counterparts, with only a fraction of the computational cost and memory usage. SwitchHead drastically reduces the number of attention matrices that have to be computed, by using MoE for the value and output projections. Our method is stable and does not need additional regularization to prevent degenerate solutions (a well-known practical issue in many existing MoE models). Our method can also be successfully combined with MoE MLP layers, to obtain "SwitchAll" where every layer of the Transformer is MoE-based, achieving a huge reduction in resource requirements.

## Acknowledgements

This research was partially funded by ERC Advanced grant no: 742870, project AlgoRNN, and by Swiss National Science Foundation grant no: 200021_192356, project NEUSYM. We are thankful for hardware donations from NVIDIA and IBM. The resources used for this work were partially provided by Swiss National Supercomputing Centre (CSCS) projects d123 and s1205.

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

# A  Appendix

## A.1  A Comment on Flash Attention

The resource reductions from Flash Attention might be, in many cases, larger than those from our method alone. However, Flash Attention depends on GPU-specific memory bandwidth/compute trade-offs, which might not be available on all hardware, especially on edge devices. SwitchHead and FlashAttention can also be combined for further speedups. We demonstrated the viability of this setup in our RoPE experiments. Additionally, certain architectures, such as shared-layer transformers, might require a drastic increase in the number of heads, which FlashAttention alone might not be able to do.

## A.2  Resource Usage of Different Methods

In this section, we discuss the compute and memory usage of different attention variants. We will define the compute in terms of the number of multiply-accumulate operations (MACs, also used by Zhang et al. [18]), which is arguably better defined than FLOPs (e.g., does one step of the matrix multiplication count as 1 FLOP or 2? Do we include the softmax?). All calculations will be presented for a single attention layer for a single sequence, and they are presented this way in all our tables. Both the memory and compute requirements scale linearly with both the batch size and the number of layers.

Consider a sequence of inputs of length $T$, with representation size $d_{\text{model}}$. Let $d_{\text{head}}$ be the width of the key, query and value projections used for the attention layer. For Transformer XL-style attention, let the size of the context be $CT$, where $C - 1$ is the number of past chunks included in the context of the current attention step. We can divide the computation into two major parts: calculating the projections, which do not involve the attention map, and calculating the attention map and projecting the sequence of values using it.

First, consider the case of the standard Transformer XL [32]. Here, from the input $\boldsymbol{x} \in \mathbb{R}^{T \times d_{\text{model}}}$, we calculate the $\boldsymbol{K}^h, \boldsymbol{Q}^h, \boldsymbol{V}^h \in \mathbb{R}^{T \times d_{\text{head}}}$ using projection matrices of shape $\mathbb{R}^{d_{\text{model}} \times d_{\text{head}}}$. The output after the attention is projected in a similar manner (Eq. 3). Thus, the projections take a total of $4T d_{\text{model}} d_{\text{head}}$ MACs per head. For backpropagation, we have to store all the intermediate results. This takes $T d_{\text{head}}$ numbers of $\boldsymbol{K}^h, \boldsymbol{Q}^h$ and $\boldsymbol{V}^h$. Also, the projected values should be stored. They have an identical shape, therefore, the total memory used by projections is $4T d_{\text{head}}$ numbers per head. Now consider the resource usage related to the attention matrix. It involves calculating the product of $\boldsymbol{Q}^h \boldsymbol{K}^{h\mathsf{T}}$, which takes $d_{\text{head}} C T^2$ MACs (multiplication by $C$ is needed because the shape of $\boldsymbol{K}^h$ and $\boldsymbol{V}^h$ for Transformer XL is $CT \times d_{\text{head}}$). The projection of the values with the attention matrix $\boldsymbol{A}^h \boldsymbol{V}^h$ is similar. For the memory usage, the attention needs $CT^2$ numbers, but it needs to be stored both before and after the activation function. In addition, calculating the projection of the position encodings is necessary. This depends on the implementation, but in our case, it involves a matrix multiplication, and the total amount of computation is $2d_{\text{head}} d_{\text{model}} TC$, and it needs $2d_{\text{head}} TC$ numbers of storage. Thus the resource requirements are:

$$N_{\text{MAC}}^{\text{XL}} = n_{\text{heads}} \left( 4T d_{\text{head}} d_{\text{model}} + 2CT^2 d_{\text{head}} + 2CT d_{\text{head}} d_{\text{model}} \right) \tag{11}$$

$$N_{\text{mem}}^{\text{XL}} = n_{\text{heads}} \left( 4T d_{\text{head}} + 2CT^2 + 2CT d_{\text{head}} \right) \tag{12}$$

The resource usage of SwitchHead is different. First, the number of heads $n_{\text{heads}}$ is significantly reduced, but $d_{\text{head}}$ is typically larger. Additionally, there are $k$ experts active at the same time. Here, we only consider the case where the value and outputs are experts, but $\boldsymbol{Q}^h$ and $\boldsymbol{K}^h$ are not (this version performs the best; see Sec. 3.1). Then, we have two projections that are identical with that of Transformer XL, and two MoE-based projections. These use $T k d_{\text{model}} d_{\text{head}}$ MACs to calculate the projection and another $T k d_{\text{head}}$ to calculate their weighted average. With a smart kernel implementation, memory usage is not affected by $k$, thus the formula remains the same as Eq. 12 (note, however, that $n_{\text{heads}}$ and $d_{\text{head}}$ are very different in practice). The compute requirement can be calculated as:

$$N_{\text{MAC}}^{\text{SwitchHead}} = n_{\text{heads}} \left( 2T d_{\text{head}} d_{\text{model}} + 2T k d_{\text{head}} (d_{\text{model}} + 1) + 2CT^2 d_{\text{head}} + 2CT d_{\text{head}} d_{\text{model}} \right) \tag{13}$$

Additionally, the expert selection logic needs minimal additional resources, which can be ignored. Note that the comparison between the MACs of the standard (Eq. 11) and SwitchHead (Eq. 13) depends on the exact values of the hyper-parameters. However, as we'll see in Sec. 3, in our typical

Table 6: Performance of SwitchHead with $E = 5$ experts and $n_{\text{heads}} = 2$ heads. Different projections are either experts or fixed for the given head. Columns V, K, Q, and O show whether the given projection is an expert. Parameter-matched baseline with $n_{\text{heads}} = 10$ and $n_{\text{heads}} = 2$ are shown. Models sorted by perplexity. 47M parameters models on Wikitext 103.

| Model | $n_{\text{heads}}$ | V | K | Q | O | Perplexity $\downarrow$ |
|---|---|---|---|---|---|---|
| SwitchHead | 2 | Y | N | N | Y | 12.27 |
| SwitchHead | 2 | N | N | N | Y | 12.30 |
| Transformer | 10 | - | - | - | - | 12.31 |
| SwitchHead | 2 | N | Y | N | Y | 12.36 |
| SwitchHead | 2 | Y | Y | N | Y | 12.37 |
| SwitchHead | 2 | Y | N | Y | Y | 12.42 |
| SwitchHead | 2 | Y | N | N | N | 12.45 |
| SwitchHead | 2 | N | N | Y | Y | 12.45 |
| SwitchHead | 2 | Y | N | Y | N | 12.51 |
| SwitchHead | 2 | Y | Y | Y | Y | 12.57 |
| SwitchHead | 2 | N | Y | Y | Y | 12.59 |
| SwitchHead | 2 | Y | Y | Y | N | 12.61 |
| SwitchHead | 2 | Y | Y | N | N | 12.69 |
| Transformer | 2 | - | - | - | - | 12.74 |
| SwitchHead | 2 | N | N | Y | N | 12.75 |
| SwitchHead | 2 | N | Y | N | N | 12.79 |
| SwitchHead | 2 | N | Y | Y | N | 12.90 |

configurations, SwitchHead provides good predictive performance with significantly lower $n_{\text{heads}}$ compared to the standard Transformer, resulting in reduced resource usage in the end.

The resource requirements of MoA [19] are very similar to those of Transformer XL , except that it uses a single shared key and value projection for each head.

$$N_{\text{MAC}}^{\text{MoA}} = (2n_{\text{heads}} + 2)T d_{\text{head}} d_{\text{model}} + 2n_{\text{heads}} CT^2 d_{\text{head}} + 2CT d_{\text{head}} d_{\text{model}} \tag{14}$$

$$N_{\text{mem}}^{\text{MoA}} = (2n_{\text{heads}} + 2)T d_{\text{head}} + 2n_{\text{heads}} CT^2 + 2CT d_{\text{head}} \tag{15}$$

## A.3 The Importance of Different Projections

In order to analyze which projections are the most important to be mixture-of-experts, we exhaustively tried all combinations. We analyze our 47M parameter models on WikiText 103 dataset. We show the results in Tab. 6. We also include a parameter-matched baseline with two heads, which serves as a lower bound for the performance. We found that the value and output projections are the most important, and having key and query projections hurts the performance. This is possible because we perform all our experiments in a parameter-matched setting. Allocating parameters to these projections uses the budget that can be otherwise spent on other parts of the network. In our preliminary experiments, we found that, allowing the parameter budget to increase, more experts always help.

## A.4 RoPE Positional Encodings

All of our experiments in the main paper have used a Transformer XL model. Thus, it remains unclear whether SwitchHead is specific to this model or can be also used with other attention methods. As an alternative, we consider RoPE positional encodings [28] without the XL cache (thus, the attention matrices are square). This is the standard setup used by modern language models, such as all versions of Llama [27]. We tested these models in Wikitext 103 and C4. The results are shown in Tab. 7, and zero-shot performance on downstream tasks in Tab. 8. This shows that SwitchHead also performs well in the standard setup and is not tied to Transformer XL.

## A.5 Hyperparameters

We train all our models with Adam optimizer [40], with a batch size of 64, a learning rate of 0.00025, and gradient clipping with a maximum norm of $\kappa$. Large models ($> 200K$ parameters) use a learning

Table 7: Perplexity of SwitchHead compared to dense baseline, using RoPE positional encoding and no XL cache. Memory usage is specified in number of floats. Models sorted by perplexity.

| Dataset | #total params | Model | $n_{\text{heads}}$ | ppl $\downarrow$ | MACs | Memory |
|---|---|---|---|---|---|---|
| Wikitext 103 | 45M | SwitchHead | 2 | 12.75 | 285.6M | 1.3M |
| | | Transformer | 10 | 12.78 | 560.9M | 6.1M |
| | | Transformer | 2 | 12.96 | 560.9M | 1.9M |
| | 244M | SwitchHead | 4 | 10.00 | 4.2G | 18.4M |
| | | Transformer | 16 | 10.17 | 6.4G | 37.7M |
| | | Transformer | 2 | 10.26 | 6.4G | 8.4M |
| C4 | 45M | SwitchHead | 2 | 23.69 | 285.6M | 1.3M |
| | | Transformer | 10 | 23.79 | 560.9M | 6.1M |
| | 244M | SwitchHead | 4 | 16.41 | 4.2G | 18.4M |
| | | Transformer | 16 | 16.35 | 6.4G | 37.7M |

Table 8: Zero-shot task performance of SwitchHead using RoPE positional encodings and no XL cache, trained on C4 dataset, compared to dense Transformer baseline with matched number of parameters.

| Model | #total params | ppl $\downarrow$ | Lambada $\uparrow$ | BLiMP $\uparrow$ | CBT $\uparrow$ |
|---|---|---|---|---|---|
| SwitchHead | 45M | 23.69 | 20.9% | 77.3% | - |
| Transformer | 45M | 23.76 | 20.3% | 73.8% | - |
| SwitchHead MAC-matched | 54M | 22.18 | 22.6% | 77.4% | - |
| SwitchHead Shared selection | 45M | 23.63 | 20.3% | 76.0% | - |
| SwitchHead | 243M | 16.41 | 30.5% | 79.9% | 83.8% |
| Transformer | 243M | 16.35 | 29.8% | 76.1% | 83.9% |
| SwitchHead MAC-matched | 314M | 15.63 | 30.5% | 80.5% | 84.6% |
| SwitchHead Shared selection | 243M | 16.59 | 28.1% | 79.1% | 83.7% |

rate warm-up of 4k steps. All models, except the SwitchAll model, use a dropout on the MLP layers, 0.1 for the small models and 0.2 for the large ones. Detailed hyperparameters are shown in the Tab. 9. $\sigma$-MoE related hyperparameters for the SwitchAll models are identical to those of Csordás et al. [17]. For Transformer XL models, we always use a single additional chunk of context, both in training and validation time. $d_{\text{head}}$ and $d_{\text{ff}}$ are derived in a systematic way, see Sec. 3 for more details.

### A.6   A Note on the Parameter Count of the SwitchAll

It can be seen in Tab. 3 that the parameter count of the SwitchAll models is often less than that of their dense counterparts. The reason is that we normally compensate for the final difference in the number of parameters by increasing $d_{\text{ff}}$ (see Sec. 3 for details of the parameter matching). However, that can only be done in a very coarse-grained way with $\sigma$-MoE: the size of all experts must be increased at once, and the CUDA kernel supports only sizes of multiple of 4. Therefore, increasing the size of the experts would add too many parameters and the model would outgrow the baseline. For this reason, we simply keep the hyperparameters for Csordás et al. [17] and combine them with our SwitchHead configuration from Tab. 2.

### A.7   Visializing all Attention Heads

As discussed in Sec. 4, we analyze the attention maps of SwitchHead and compare them with the dense models. We show all the attention maps of the models trained on ListOps in Fig. 3 and Fig. 3. We show individual heads of SwitchHead, including the expert selection scores in Fig. 5. Some selected attention maps of our 47M parameter models on Wikitext 103 are shown in Fig. 6.

### A.8   Compute Requirements

We report the compute used for our experiments, including the GPU type, count (the number of GPUs used per experiment, and not the total in the machine), and the runtime in "hh:mm" format

Table 9: Hyperparameters used for our models.

| Model | Dataset | $n_{\text{heads}}$ | #params | $d_{\text{head}}$ | $d_{\text{ff}}$ | E | k | T | $n_{\text{layers}}$ | $\kappa$ |
|---|---|---|---|---|---|---|---|---|---|---|
| SwitchHead | | 2 | 47M | 76 | 2080 | 5 | 3 | 256 | 16 | 0.1 |
| Transformer | C4 | 10 | 47M | 41 | 2053 | - | - | 256 | 16 | 0.1 |
| Transformer | | 2 | 47M | 205 | 2053 | - | - | 256 | 16 | 0.1 |
| SwitchHead | | 4 | 262M | 112 | 4188 | 4 | 2 | 512 | 18 | 0.25 |
| Transformer | C4 | 16 | 262M | 64 | 4110 | - | - | 512 | 18 | 0.25 |
| Transformer | | 4 | 262M | 256 | 4110 | - | - | 512 | 18 | 0.25 |
| SwitchHead | | 2 | 47M | 76 | 2080 | 5 | 2 | 256 | 16 | 0.1 |
| Transformer | Wikitext 103 | 10 | 47M | 41 | 2053 | - | - | 256 | 16 | 0.1 |
| Transformer | | 2 | 47M | 205 | 2053 | - | - | 256 | 16 | 0.1 |
| SwitchHead | | 2 | 262M | 132 | 4147 | 8 | 4 | 512 | 18 | 0.25 |
| Transformer | Wikitext 103 | 16 | 262M | 64 | 4110 | - | - | 512 | 18 | 0.25 |
| Transformer | | 2 | 262M | 512 | 4110 | - | - | 512 | 18 | 0.25 |
| SwitchHead | | 2 | 47M | 76 | 2080 | 5 | 3 | 256 | 16 | 0.1 |
| Transformer | peS2o | 10 | 47M | 41 | 2053 | - | - | 256 | 16 | 0.1 |
| Transformer | | 2 | 47M | 205 | 2053 | - | - | 256 | 16 | 0.1 |
| SwitchHead | | 4 | 262M | 112 | 4188 | 4 | 2 | 512 | 18 | 0.25 |
| Transformer | peS2o | 16 | 262M | 64 | 4110 | - | - | 512 | 18 | 0.25 |
| Transformer | | 4 | 262M | 256 | 4110 | - | - | 512 | 18 | 0.25 |
| SwitchHead | | 2 | 41M | 112 | 2088 | 4 | 2 | 512 | 12 | 0.25 |
| Transformer | Enwik8 | 8 | 41M | 64 | 2053 | - | - | 512 | 12 | 0.25 |
| Transformer | | 2 | 41M | 256 | 2053 | - | - | 512 | 12 | 0.25 |
| SwitchHead (RoPE) | Wikitext 103 | 2 | 45M | 64 | 2092 | 5 | 3 | 512 | 16 | 0.1 |
| Transformer (RoPE) | | 10 | 45M | 41 | 2053 | - | - | 512 | 16 | 0.1 |
| SwitchHead (RoPE) | Wikitext 103 | 4 | 243M | 100 | 4136 | 4 | 2 | 1024 | 18 | 0.25 |
| Transformer (RoPE) | | 16 | 244M | 64 | 4110 | - | - | 1024 | 18 | 0.25 |
| SwitchAll | Wikitext 103 | 2 | 47M | 76 | 1648 | 5 | 2 | 256 | 16 | 0.25 |
| SwitchAll | Wikitext 103 | 4 | 259M | 112 | 4096 | 4 | 2 | 512 | 18 | 0.25 |
| SwitchAll | C4 | 2 | 47M | 76 | 1648 | 5 | 3 | 256 | 16 | 0.25 |
| SwitchAll | C4 | 4 | 259M | 112 | 4096 | 4 | 2 | 512 | 18 | 0.25 |
| SwitchAll | peS2o | 2 | 47M | 76 | 1648 | 5 | 3 | 256 | 16 | 0.25 |
| SwitchAll | peS2o | 4 | 259M | 112 | 4096 | 4 | 2 | 512 | 18 | 0.25 |

in Tab. 10. We report the total number of CPUs ($N_{\text{CPU}}$) and RAM because they are shared between concurrent runs. Note that most of the experiments were done prior to the much faster, Triton-based kernel implementation. Because of this, the runtimes appear longer for SwitcHead compared to the baseline. For timing benchmarks with our new kernel, see Tab. 5.

Note that we only report the resources used for the paper here. We estimate that the total cost of the failed experiments and preliminary runs is around 10 times higher than this.

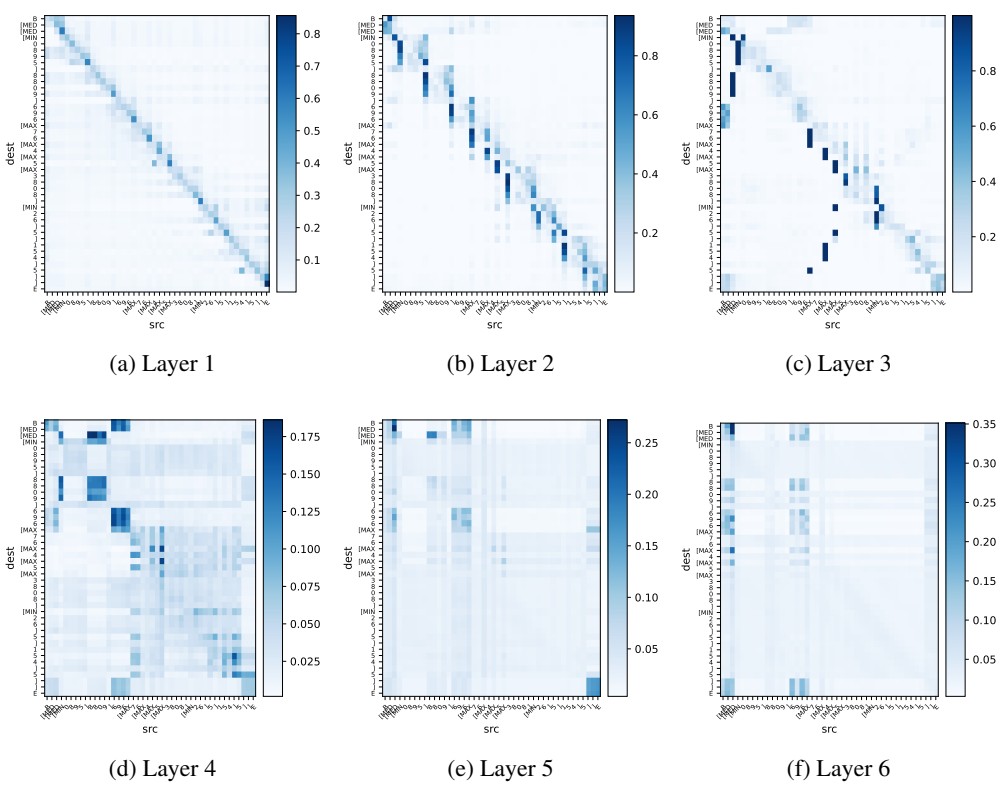

(a) Layer 1          (b) Layer 2          (c) Layer 3

(d) Layer 4          (e) Layer 5          (f) Layer 6

Figure 3: The maximum of all attention maps for a SwitchHead model on ListOps.

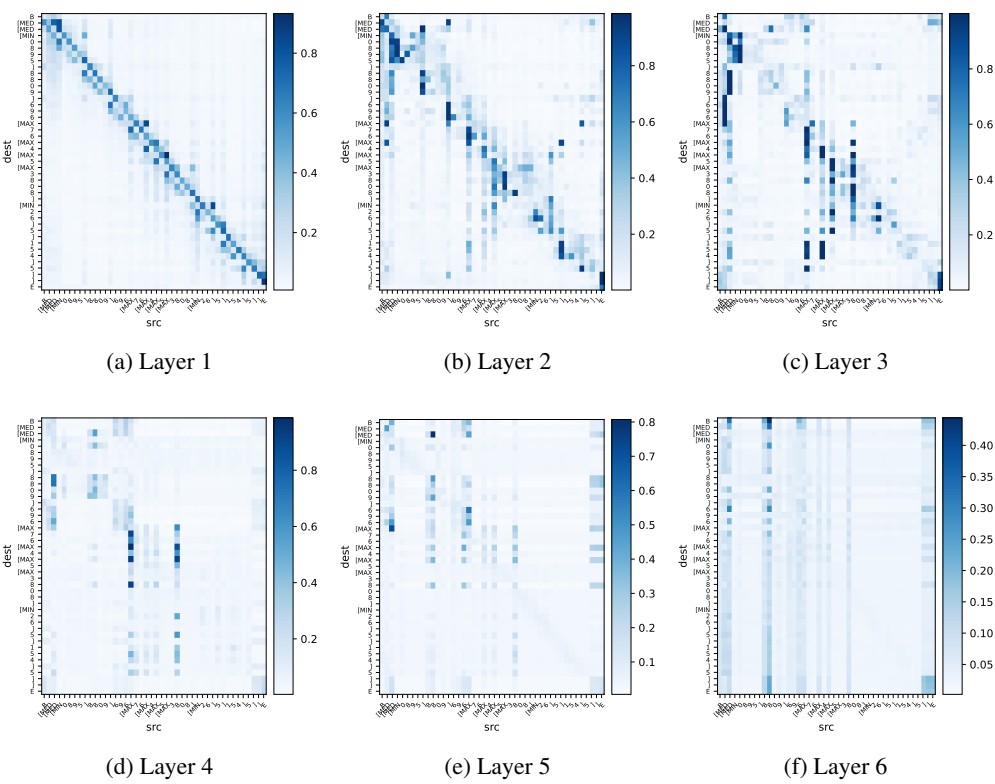

(a) Layer 1    (b) Layer 2    (c) Layer 3

(d) Layer 4    (e) Layer 5    (f) Layer 6

Figure 4: The maximum of all attention maps for a standard Transformer model on ListOps.

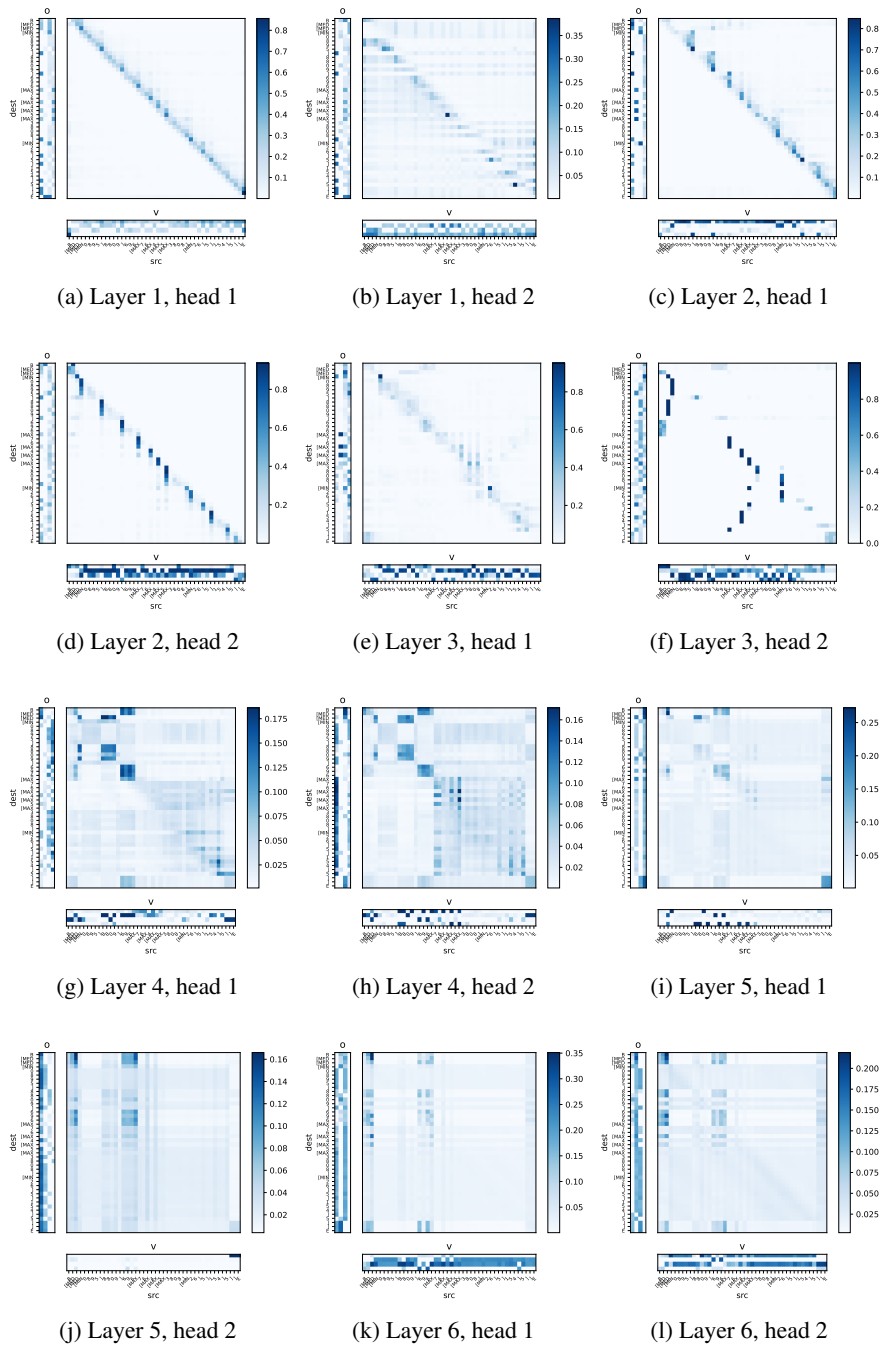

Figure 5: Details for individual heads of the SwitchHead model on ListOps. On the left side of each attention plot, the selection of the output projection expert is shown. Similarly, at the bottom, the selection of the value projection selection is visible. In the selection maps, dark blue always corresponds to 1, while white is 0. The adaptive scale shown to the right of the attention map is for the map only.

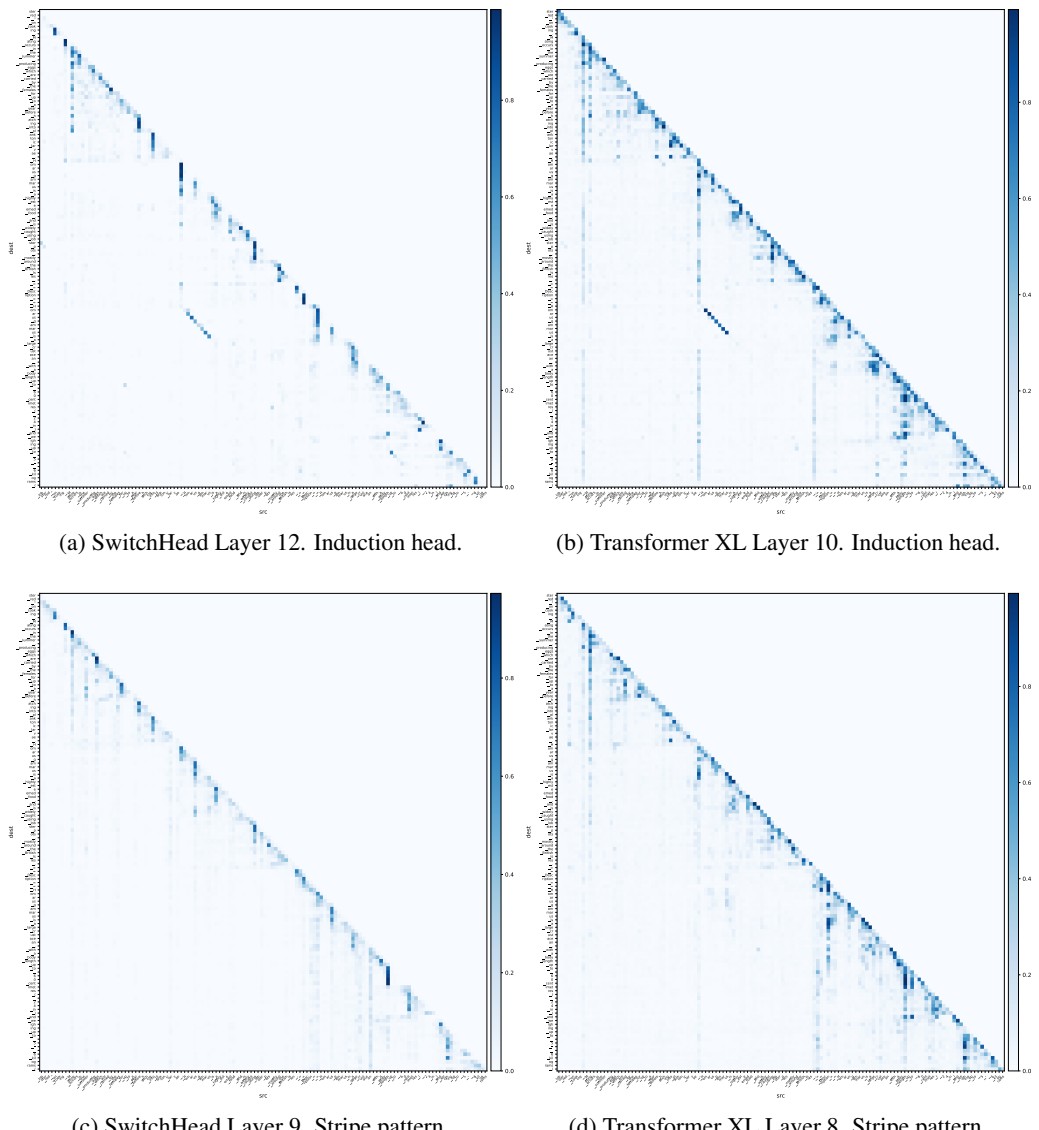

(a) SwitchHead Layer 12. Induction head.

(b) Transformer XL Layer 10. Induction head.

(c) SwitchHead Layer 9. Stripe pattern.

(d) Transformer XL Layer 8. Stripe pattern.

Figure 6: Induction head copying the rare name "Homarus" in (a) SwitchHead and (b) Transformer XL baseline. The attention matrix is square because it is the first chunk of the sequence, without any extra context. Typical vertical line pattern in (c) SwitchHead and (b) Transformer XL baseline.

Table 10: Training hardware information for the experiments reported in the paper

| Model | #params | Dataset | $G$ | GPU Type | $N_{GPU}$ | $N_{CPU}$ | RAM | Duration |
|---|---|---|---|---|---|---|---|---|
| SwitchAll | 259M | C4 | 4 | V100-32GB-LS | 8 | 40 | 503G | 24:06 |
| SwitchAll | 259M | peS2o | 4 | V100-32GB-LS | 8 | 40 | 503G | 30:00 |
| SwitchAll | 259M | Wikitext 103 | 4 | RTX 4090 | 4 | 24 | 251G | 22:58 |
| SwitchAll | 47M | C4 | 2 | RTX 3090 | 1 | 24 | 220G | 22:14 |
| SwitchAll | 47M | peS2o | 2 | RTX 3090 | 1 | 24 | 220G | 22:49 |
| SwitchAll | 47M | Wikitext 103 | 2 | RTX 3090 | 1 | 24 | 251G | 6:03 |
| SwitchHead | 243M | Wikitext 103 | 4 | V100-32GB | 4 | 40 | 503G | 147:09 |
| SwitchHead | 262M | C4 | 4 | V100-32GB-LS | 8 | 40 | 503G | 26:38 |
| SwitchHead | 262M | peS2o | 4 | V100-32GB-LS | 8 | 40 | 503G | 27:43 |
| SwitchHead | 262M | Wikitext 103 | 2 | V100-32GB | 4 | 40 | 503G | 31:42 |
| SwitchHead | 41M | Enwik8 | 2 | V100-32GB | 1 | 40 | 503G | 13:45 |
| SwitchHead | 45M | Wikitext 103 | 2 | RTX 3090 | 1 | 24 | 251G | 17:28 |
| SwitchHead | 47M | C4 | 2 | V100-32GB | 1 | 40 | 503G | 15:36 |
| SwitchHead | 47M | peS2o | 2 | V100-32GB | 1 | 40 | 503G | 16:17 |
| SwitchHead | 47M | Wikitext 103 | 2 | RTX 3090 | 1 | 24 | 251G | 13:09 |
| Transformer | 262M | C4 | 4 | V100-32GB | 8 | 40 | 503G | 11:55 |
| Transformer | 262M | C4 | 16 | V100-32GB-LS | 8 | 40 | 503G | 20:21 |
| Transformer | 262M | peS2o | 4 | V100-32GB | 8 | 40 | 503G | 17:08 |
| Transformer | 262M | peS2o | 16 | V100-32GB-LS | 8 | 40 | 503G | 25:56 |
| Transformer | 262M | Wikitext 103 | 2 | P100-16GB | 8 | 12 | 62G | 0:00 |
| Transformer | 262M | Wikitext 103 | 16 | A100-80GB | 2 | 64 | 503G | 31:51 |
| Transformer | 41M | Enwik8 | 2 | RTX 3090 | 1 | 24 | 220G | 15:38 |
| Transformer | 41M | Enwik8 | 8 | V100-32GB-LS | 2 | 40 | 503G | 16:04 |
| Transformer | 47M | C4 | 2 | V100-32GB | 1 | 40 | 503G | 10:29 |
| Transformer | 47M | C4 | 10 | V100-32GB | 1 | 40 | 503G | 16:57 |
| Transformer | 47M | peS2o | 2 | V100-32GB | 1 | 40 | 503G | 11:07 |
| Transformer | 47M | peS2o | 10 | V100-32GB | 1 | 40 | 503G | 17:55 |
| Transformer | 47M | Wikitext 103 | 2 | V100-32GB | 1 | 40 | 503G | 10:06 |
| Transformer | 47M | Wikitext 103 | 10 | V100-32GB | 1 | 40 | 503G | 18:51 |
| Transformer (RoPE) | 244M | Wikitext 103 | 16 | RTX 3090 | 4 | 24 | 251G | 30:30 |
| Transformer (RoPE) | 45M | Wikitext 103 | 10 | V100-32GB | 1 | 40 | 503G | 15:30 |

