# OpenReview forum: "SwitchHead: Accelerating Transformers with Mixture-of-Experts Attention"
_NeurIPS.cc/2024/Conference — NeurIPS 2024 poster_

### Official Review · Reviewer_Q7cF · 2024-07-03

**Soundness:** 3
**Presentation:** 3
**Contribution:** 3
**Rating:** 6
**Confidence:** 4

**Summary:**

This paper introduces SwitchHead, a novel MoE architecture for the attention layer. Unlike the existing MoA approach, which computes the output of the attention layer as a weighted average of the top-k heads determined by a learnable routing mechanism, SwitchHead independently applies expert mixtures to the heads' key, value, and output projections. This design allows SwitchHead to achieve comparable or superior performance to MoA and dense Transformers while requiring less computational power.  The reduced compute cost is achieved by performing fewer matrix multiplications: since fewer heads need to be instantiated, the attention matrix operations are performed less frequently (same number of params is ensured by setting d_head accordingly). Reduced memory cost is achieved by having to store less activations for the backward pass. The paper points out that only applying MoE to value and output projection in each head performs sufficiently well as compared to applying MoE to the key and query projections as well.

**Strengths:**

Overall, this is a well-crafted paper that presents a straightforward yet effective method for efficiently integrating MoE layers into self-attention blocks.

Originality: good. The idea of implementing of transforming self-attention into an MoE is not new, yet the specific method presented and analyzed here is novel enough.

Quality: good. The claims are well supported by evidences. Experiments are well designed and seem to be sufficient overall.

Clarity: the paper is well organized. In terms of writing, some formulations require further clarifications (see weaknesses/questions)

Significance: good. The paper makes a moderate contribution to the field and can potentially impacting future research.

**Weaknesses:**

I cannot identify any fundamental weaknesses with the paper beyond several confusing passages that might require further clarification, and a couple of potentially missing citations (see questions).

**Questions:**

- similar techniques for transforming attention heads into MoE have been proposed in the context of parameter efficient MoEs for the fine-tuning stage where experts are represented with LoRA adapters, authors can consider mentioning these works [2,3,4 inter alia]
- ll. 88 - 91: "Intuitively, the above method corresponds to choosing a subset of attention heads based on the destination side alone": given that routing vector s is calculated conditioned on the input x, I do not understand why is it the case that attention heads are selected based on the destination (queries and outputs) side alone. In my understanding the attention head selection is dependent on the input x.
- ll. 93 - 96 (related to above point) the explanation is somewhat confusing, why is it the case that in the worst case all possible source projections have to be computed?
- l. 439: does the 4 in the calculation of the total MACs for the projections already include the output projection?
- ll. 158 - 161: it might be worth noting that the idea of sharing key and value projection across heads has been introduced in earlier work on multi-query attention [1]. Additionally, to enhance the overall clarity, it could be useful to explicitly highlight how exactly MoA differs from the naive attention head mixing technique described in the first part of the Sec. 2.2.
- ll. 119 - 120: it would be helpful if authors could provide a brief discussion clarifying how and why exactly parameter-matched setting "better reflects the task of language modelling"?
- ll. 225 - 227: the visualization in Fig. 2 is only for 1 layer, and not for all layers as stated in the text
- ll. 282 - 287: does not exactly fit into the "Limitations" section, since, according to the authors (ll. 266-269) FlashAttention is orthogonal to SwitchHead
- how important is it to have MoE in the attention the the fully-connected block is already an MoE? Could authors add comparison to σ-MoE?

Overall solid paper, I am happy to revisit my score upon the rebuttal.

[1] Shazeer, Noam. "Fast transformer decoding: One write-head is all you need." arXiv preprint arXiv:1911.02150 (2019).

[2] Page-Caccia, Lucas, et al. "Multi-head adapter routing for cross-task generalization." Advances in Neural Information Processing Systems 36 (2024).

[3] Zadouri, Ted, et al. "Pushing mixture of experts to the limit: Extremely parameter efficient moe for instruction tuning." arXiv preprint arXiv:2309.05444 (2023).

[4] Ostapenko, Oleksiy, et al. "Towards modular llms by building and reusing a library of loras." arXiv preprint arXiv:2405.11157 (2024).

**Limitations:**

Limitations are addressed in section 6.

---

> ### Author Rebuttal · Authors · 2024-08-06
>
> We would like to thank the reviewer for their insightful review and for positive comments on the clarity and methodology of the paper. Please find our responses as follows:
>
> > similar techniques for transforming attention heads into MoE have been proposed in … authors can consider mentioning these works [2,3,4 inter alia]
>
> We would like to thank the reviewer for pointing this out. We will include the discussion of these papers in the final version.
>
> > given that routing vector s is calculated conditioned on the input x, I do not understand why is it the case that attention heads are selected based on the destination (queries and outputs) side alone. …
>
> We agree that at this spot, this description is not very clear (this becomes hopefully clearer later, in light of our description starting at L97 until Eq. 10, that introduces MoE everywhere, in both source and destination sides). The reviewer is right to point out that everything is a function of the input x, including the output of the routing functions. In self-attention, all of key, value, query vectors are indeed computed from the input x for each position. But once we have these vectors, from the viewpoint of attention computation, the current query defines the destination, and keys and values define the source. In the MoE version of attention, we allocate a routing function for each of key/value/query projection; these routing functions belong to the source or destination side accordingly. Now if we compare Eq 10 and Eq 6, one can notice that the routing function in Eq. 6 effectively corresponds to what we define as the destination-side routing in Eq 10. We will improve the clarity of this passage in the final version. Thank you for pointing this out.
>
> > the explanation is somewhat confusing, why is it the case that in the worst case all possible source projections have to be computed?
>
> Considering the ‘row’ of an attention matrix as ‘destination’ and ‘column’ as 'source’, routing on the “destination side” means that we only select/use K experts to compute a row. However, different rows can still decide to use different selections of these K experts. This means that in practice, there are more than K active experts per column. In the worst case, i.e., if all rows decide to use different K experts, this requires K times the number of row active experts, or simply the total number of experts (if such a product exceeds the total number of experts, which is typically the case as there are more rows than the number of experts). This requires all possible key and value projections to be computed even if only a subset is used in each row. Again, we agree with the reviewer that this is only implicit and confusing in the current description. We will improve this in the final version. Thank you very much for pointing this out.
>
> > does the 4 in the calculation of the total MACs for the projections already include the output projection?
>
> Yes, it does. We will clarify this in the final version.
>
> > it might be worth noting that the idea of sharing key and value projection across heads has been introduced in earlier work on multi-query attention [1].
>
> We would like to thank the reviewer for pointing this out. We will include a discussion of [1] in the final version.
>
> > Additionally, to enhance the overall clarity, it could be useful to explicitly highlight how exactly MoA differs from the naive attention head mixing technique described in the first part of the Sec. 2.2.
>
> We would like to thank the reviewer for pointing this out. The main difference is that MoA fixes the number of K and V projections to 1, thus the efficiency limitations we describe do not apply to it. However, this limits its expressibility. Despite having a single key and value projections, it computes K different attention maps.
>
> > it would be helpful if authors could provide a brief discussion clarifying how and why exactly parameter-matched setting "better reflects the task of language modelling"?
>
> The parameter-matched setting is crucial to evaluate the model’s *expressiveness* in the LLM tasks where the number of parameters has a high impact on the model performance. We consider this setting to be particularly important to evaluate the true expressiveness of MoEs compared to their dense counterparts.
>
> Please note that we also provide additional results in a MAC-matched setting in Tab 4.
>
> > does not exactly fit into the "Limitations" section, since, according to the authors (ll. 266-269) FlashAttention is orthogonal to SwitchHead
>
> We agree with the reviewer that this paragraph does not fit here. We’ll move this to Appendix A. Thank you for pointing this out.
>
> > the visualization in Fig. 2 is only for 1 layer, and not for all layers as stated in the text
> We would like to thank the reviewer for pointing this out. We will fix this in the final version.
>
> > how important is it to have MoE in the attention the the fully-connected block is already an MoE? Could authors add comparison to σ-MoE?
>
> The effects of σ-MoE and SwitchHead are orthogonal to each other, as they affect independent parts of the network. In the experimental setting of the paper, they are both set up to result in the most significant speedup with a marginal perplexity gain. Thus, by construction, combining them would never result in an increased perplexity. Speed-wise, with the current implementation, σ-MoE does not provide wall-clock gains on this model scale, in contrast to SwitchHead which provides a significant wall-clock speedup (see Tab 5). It saves, however, around 1.5Gb of additional memory compared to SwitchHead alone in the 262M param setting of Tab 5. With a higher d_model, the savings are significantly more substantial.
>
>
> We believe our response above resolves all the concerns that the reviewer has raised.
> These questions are extremely valuable for us and will enable us to improve our paper.
> If the reviewer finds our response useful, please consider increasing the score. Thank you very much.

---

> > ### Comment · Reviewer_Q7cF · 2024-08-09
> >
> > I would like to thank the authors for their responses. Upon clarification, I now understand a bit better what the authors mean in ll. 84-96. It would help future readers if authors could incorporate the clarifications regarding these lines in the future paper version.

---

> > > ### Author Response · Authors · 2024-08-09
> > >
> > > Thank you very much for your response! We are glad to hear that the reviewer found our response useful! Yes, we will make sure to include these clarifications in the next version. Thank you again for this valuable feedback!

---

### Official Review · Reviewer_YarL · 2024-07-08

**Soundness:** 3
**Presentation:** 3
**Contribution:** 3
**Rating:** 6
**Confidence:** 4

**Summary:**

This paper introduces SwitchHead, a Mixture of Experts (MoE) method for improving the efficiency of the self-attention layer in Transformers. Unlike traditional MoE methods focused mainly on feedforward layers, SwitchHead effectively reduces both compute and memory requirements, achieving significant wall-clock speed improvements without compromising language modeling performance. This is accomplished by reducing the number of attention matrices needed, offering substantial savings in computational resources, which is validated through extensive experiments across multiple datasets and model sizes.

**Strengths:**

The paper is well-written and easy to follow. The proposed method demonstrates substantial improvements in computational efficiency by reducing the number of attention matrices needed, which leads to lower compute and memory usage compared to standard Transformers.

**Weaknesses:**

The proposed SwitchHead method shares similarities with the Mixture of Attention Heads (MoA), but a key distinction lies in its use of a non-competitive activation function (sigmoid) instead of SoftMax. This design choice is important to understand, and I would appreciate an explanation for opting for the sigmoid activation function over SoftMax in this context.

**Questions:**

1. The authors mention employing a parameter-matched setting to better reflect the task of language modeling. However, the paper lacks explicit justification for why this setting is more representative of language modeling tasks. Additionally, the description of parameters in Table 1 is ambiguous—it should be clarified whether the numbers represent total parameters or only those that are actively used during inference.

2. The paper occasionally uses notations without adequate explanations, which can lead to confusion. For instance, the term “Shared selection” used in Table 4 is not clearly defined.

3. In Table 5, the authors present the training times for the baseline Transformer and SwitchHead models, but the training time for the MoA model is not included. To provide a comprehensive comparison and better evaluate the efficiency of SwitchHead relative to other models, it would be beneficial if the authors could also include the training time data for the MoA model.

**Limitations:**

The authors have discussed the limitations of their study.

---

> ### Author Rebuttal · Authors · 2024-08-06
>
> We would like to thank the reviewer for their insightful review and for positive comments on the clarity of the paper. Please find our responses as follows:
>
> > A key distinction lies in its use of a non-competitive activation function (sigmoid) instead of SoftMax. This design choice is important to understand, and I would appreciate an explanation for opting for the sigmoid activation function over SoftMax in this context.
>
> We would like to emphasize that the differences between MoA and SwitcHead are more than just the activation function. SwitchHead also uses multiple heads, while MoA uses a single Q and K projection. Moreover, MoA does the weighted average after the attention matrix computation, which makes it slower. Section 3.2 lists all these differences.
>
> The use of sigmoid activation function in MoE was introduced by σ-MoE [1] and intuitively motivated by similarity to an approximate feedforward layer. Moreover, the authors of [2] provide a detailed theoretical analysis of the sigmoid activation function for MoEs, and they show that it converges faster.
>
> > The authors mention employing a parameter-matched setting to better reflect the task of language modeling. However, the paper lacks explicit justification for why this setting is more representative of language modeling tasks.
>
> The parameter-matched setting is crucial to evaluate the model’s *expressiveness* in the LLM tasks where the number of parameters has a high impact on the model performance. We consider this setting to be particularly important to evaluate the true expressiveness of MoEs compared to their dense counterparts.
>
> While the compute-matched setup has values when considering certain practical settings, it gives an “unfair” advantage to MoEs in terms of comparison, as we can easily add extra parameters to an MoE without significantly increasing compute requirements. Here we wanted to show that our SwitchHead is capable, even without considering such an advantage, by evaluating its pure expressiveness in the more challenging parameter-matched setting.
>
> Please note that we also provide additional results in a MAC-matched setting in Tab 4.
>
> > Additionally, the description of parameters in Table 1 is ambiguous—it should be clarified whether the numbers represent total parameters or only those that are actively used during inference.
>
> We would thank the reviewer for pointing this out. It is the total number of parameters. We will clarify this in the final version.
>
> > The paper occasionally uses notations without adequate explanations, which can lead to confusion. For instance, the term “Shared selection” used in Table 4 is not clearly defined.
>
> Thank you very much for pointing this out. Shared selection refers to the case where we tie the V and O projections' selections to be identical. This marginally reduces compute and memory requirements at a slight performance cost. We agree this was confusing, we will provide the details in the final version.
>
> > In Table 5, the authors present the training times for the baseline Transformer and SwitchHead models, but the training time for the MoA model is not included. To provide a comprehensive comparison and better evaluate the efficiency of SwitchHead relative to other models, it would be beneficial if the authors could also include the training time data for the MoA model.
>
> This is an excellent suggestion. Thank you for pointing it out. For measuring the resource usage of MoA, we chose the fastest MoA model that can match the performance of the dense baseline, or simply the best MoA model when no MoA model can match the baseline performance. This resulted in choosing MoA with H=4 for the 47M model and MoA with H=8 for the 262M parameter model. Please find the updated Table 5 below. It can be seen that SwitchHead outperforms MoA in both memory usage and runtime in both cases.
>
> | Size | Model | ms/iteration | Rel. iter. time | RAM/GPU | Rel. Mem. | #GPUs | GPU type |
> | --- | --- | --- | --- | --- | --- | --- | --- |
> | 47M | Transformer | 473ms/iter | 1.0 | 20.5G | 1.0 | 1 | RTX 3090 |
> | | SwitchHead | 342ms/iter | 0.72 | 13.5G | 0.65 |  | |
> | | MoA | 412ms/iter | 0.87 | 15.3G | 0.75 | ||
> 262M | Transformer| 670ms/iter | 1.0 | 20.5G | 1.0 | 8 | V100|
> | | SwitchHead |  442ms/iter | 0.65 | 12.5G | 0.61 |  | |
> | | MoA |  851ms/iter | 1.27 | 16.4G | 0.80 |  | |
>
> We believe our response above resolves all the concerns that the reviewer has raised. If the reviewer finds our response useful, please consider increasing the score. Thank you very much.
>
> [1] Csordás et al. EMNLP 2023 (Findings). Approximating Two-Layer Feedforward Networks for Efficient Transformers.
>
> [2] Nguyen et al. Arxiv 2024. Sigmoid Gating is More Sample Efficient than Softmax Gating in Mixture of Experts.

---

> > ### Comment · Reviewer_YarL · 2024-08-10
> > **Official Comment by Reviewer YarL**
> >
> > Thank you to the authors for their comprehensive response to the review comments. The efforts made to address the concerns and clarify the points discussed are highly appreciated. Given these responses, I am inclined to increase my score.

---

> > > ### Author Response · Authors · 2024-08-11
> > >
> > > Thank you very much for the increased score! We are glad to hear that the reviewer found our response useful! Thank you again for your valuable feedback!

---

### Official Review · Reviewer_HTZS · 2024-07-12

**Soundness:** 3
**Presentation:** 3
**Contribution:** 3
**Rating:** 5
**Confidence:** 4

**Summary:**

This paper presents SwitchHead, a Mixture of Experts (MoE) method applied to the self-attention layer in transformer blocks. By applying MoE to the self-attention layer, SwitchHead reduces computational and memory costs while maintaining language modeling performance comparable to traditional dense models. It can be combined with existing MoE methods for feedforward layers, resulting in a fully MoE-based transformer, SwitchAll transformers. Tests on various language modeling datasets (C4, Enwik8, peS2o) show that SwitchHead performs well compared to models with parameter-matched settings.

**Strengths:**

- The proposed MoE method applied to self-attention layer reduces computational and memory requirements while preserving language modeling performance of dense baselines.
- The authors make effort to provide a fair comparison between the baselines and the proposed model in parameter-matched settings. They also include both the MAC and wall-clock speedup comparisons, clearly demonstrating the efficiency of the proposed method.
- The author performs extensive ablation studies to show that the proposed method performs well with different types of models and datasets. They also provide the hyperparameters used for each experiment, which helps in reproducing the benchmark results.

**Weaknesses:**

- The paper only evaluates the proposed method on language modeling datasets, lacking demonstration of its effectiveness on other important NLP tasks such as document summarization or open-domain question answering.
- While the authors claim similarity between SwitchHead and dense baseline attention maps, the provided figures suggest simplification in SwitchHead's maps. The lack of quantitative analysis (e.g., entropy measurements) weakens this claim.
- There's a concern that the simplified attention maps produced by SwitchHead could lead to performance degradation when SwitchHead used in encoder-based models due to information bottleneck, which isn't addressed in the current evaluation.
- The paper doesn't clearly explain why regularization methods used in σ-MoE (e.g., entropy maximization, expert dropout) are unnecessary for SwitchHead, potentially leaving gaps in the method's theoretical foundation.
- The paper suggests that Top-K selection can be treated as a hyperparameter, but doesn't provide a clear analysis of the method's sensitivity to different K values, leaving questions about the necessity of hyperparameter search.
- The paper mentions "preliminary experiments" multiple times without proper citation or section indicators, which may confuse readers and reduce the reproducibility of the work.

**Questions:**

- While the paper demonstrates strong performance on language modeling tasks, it would be beneficial to see results on a broader range of NLP tasks. How does SwitchHead perform on tasks such as document summarization[1] or open-domain question answering[2]? This would help demonstrate the method's versatility and potential impact beyond language modeling.
- The paper claims that attention maps from SwitchHead and dense baseline models (Transformer XL) are qualitatively similar. However, Figures 2 and 6 suggest that SwitchHead produces simplified attention maps. Could you provide quantitative analysis (e.g., entropy measurements) to more rigorously demonstrate the complexity and quality of SwitchHead's attention maps compared to the dense baseline?
- If SwitchHead indeed produces simplified attention maps, there's a concern about potential information bottleneck in encoder-based models. Have you considered evaluating SwitchHead's performance when applied to the query encoder in a retrieval-augmented generation (RAG) setup for open-domain QA tasks[2]? This could help address concerns about information loss in more complex, multi-step tasks.
- The paper mentions that SwitchHead doesn't require the extra regularization techniques used in σ-MoE, such as entropy maximization for load balancing or expert dropout. Could you clarify what specific differences between σ-MoE and SwitchHead make these extra tricks unnecessary? A more detailed explanation of this point would strengthen the theoretical foundation of your approach.
- You demonstrate that K in Top-K selection can be treated as a hyperparameter in the proposed method. How sensitive is the model's performance to different K values? If the sensitivity is low, is extensive hyperparameter searching always necessary, or could a default value be recommended for most use cases?
- The paper mentions "Our preliminary experiments" several times (e.g., lines 91, 473) without providing citations or section indicators. Could you clarify these references to improve the paper's clarity and reproducibility? Consider either expanding on these preliminary experiments in the main text or including them in an appendix.

[1] Nallapati, Ramesh, et al. "Abstractive text summarization using sequence-to-sequence rnns and beyond." arXiv preprint arXiv:1602.06023 (2016).

[2] Kwiatkowski, Tom, et al. "Natural questions: a benchmark for question answering research." Transactions of the Association for Computational Linguistics 7 (2019): 453-466.

**Limitations:**

- In section 6 and appendix A.1, the authors appropriately present the limitations and societal impacts of their work. This acknowledgment of their model framework's constraints effectively strengthens the motivation for additional research topics in future work. The authors' awareness of these limitations demonstrates a commendable level of scientific rigor and transparency. However, the claim that performance can potentially reach 80-90% raises some questions. While this projection is intriguing, it would be more convincing if supported by more robust quantitative assessments. The addition of more detailed quantitative evaluations could provide valuable insights and significantly aid future research efforts. For instance, a breakdown of current performance bottlenecks and a roadmap for potential optimizations would offer clearer guidance for researchers looking to build upon this work. Furthermore, a more granular analysis of the trade-offs between model size, computational resources, and performance could enhance the paper's contribution to the field. Overall, while the authors have done a commendable job in addressing limitations and societal impacts, the inclusion of more quantitative metrics and projections would further solidify the paper's value as a foundation for future research in this area.

---

> ### Author Rebuttal · Authors · 2024-08-06
>
> We would like to thank the reviewer for their insightful review. Please find our responses as follows:
>
> > … it would be beneficial to see results on a broader range of NLP tasks. How does SwitchHead perform on tasks such as document summarization[1] or open-domain question answering[2]?
>
> > Have you considered evaluating SwitchHead's performance when applied to the query encoder in a retrieval-augmented generation (RAG) setup for open-domain QA tasks[2]?
>
> We agree with the reviewers that naturally, more experiments on other tasks will strengthen our work. That said, with our modest compute resources, it would be hard for us to perform all of these experiments. We restricted ourselves to LM tasks since they are of central importance today, and many of the other tasks, such as QA, can be casted as LM. We would like to note that we already conducted experiments on a list of LM tasks (enwik8, WT 103, C4, peS2o) on two different scales (47M, 262M) with two different positional encodings (Transformer, RoPE) and evaluated them zero-shot on Lambada, CBT, and BLiMP. Currently, we are not advocating using our method for tasks other than self-attention in LM, although we believe it would have similar properties on those as well.
>
> > While the authors claim similarity between SwitchHead and dense baseline attention maps, the provided figures suggest simplification in SwitchHead's maps. The lack of quantitative analysis (e.g., entropy measurements) weakens this claim.
>
> > Could you provide quantitative analysis (e.g., entropy measurements) to more rigorously demonstrate the complexity and quality of SwitchHead's attention maps compared to the dense baseline?
>
> While we agree with the reviewer that comparing attention maps is an interesting topic, we are unaware of any quantitative analysis that can be indicative of a “quality of an attention map”, except the downstream performance of the model, which we measure on a wide variety of tasks and datasets. For example, it is hard to draw conclusions from raw entropy measurements as we could argue on both sides: high entropy attention maps integrate more information and distribute gradients to more tokens, potentially providing a better learning signal. On the contrary, they also “blur together” too much information, making it hard to filter out irrelevant tokens and sharply focus on the relevant ones. High entropy attention maps might be a mere byproduct of some heads being unused by the model (by setting its contribution low through V and O projections, which do not show up in the attention maps).
> We would also like to note that on algorithmic tasks, such as ListOps shown in Fig. 2, typically low-entropy, sharp, and simple attention maps work better, as these require exact focus on specific numbers and operations. Therefore, our current conclusion is to rely on the downstream performance of the model, assuming that models, including their attention maps, have to have good behavior to achieve good performance.
>
> > The paper mentions that SwitchHead doesn't require the extra regularization techniques used in σ-MoE, such as entropy maximization for load balancing or expert dropout. Could you clarify what specific differences between σ-MoE and SwitchHead make these extra tricks unnecessary?
>
>
> This is a very interesting open question. As this is an empirical observation in our experiments, we have no theoretical explanations. In fact, the situation is the same for the general theoretical studies on MoE; the current literature lacks clear explanations on when exactly regularization is necessary in MoE models. Also, the direct comparison between sigma-MoE and Switchhead is not straightforward as one is selecting slices of two layers in an MLP simultaneously, and the other one selecting individual projections in the attention.
>
> > ... How sensitive is the model's performance to different K values? If the sensitivity is low, is extensive hyperparameter searching always necessary, or could a default value be recommended for most use cases?
>
> > … a more granular analysis of the trade-offs between model size, computational resources, and performance could enhance the paper's contribution to the field.
>
> In paragraph 2 of Sec 3 (L124-131), we provide the details of our algorithm for selecting K. K=2 usually works well enough. Generally, increasing K always helps, but it makes the model slower and uses more memory since more projections have to be computed. Thus, the choice of K is a tradeoff. We chose the minimal K that matches or slightly outperforms the equivalent dense model. Increasing d_head and the number of heads have a similar effect.
>
> Generally, we do agree with the reviewer that a more extensive hyperparameter analysis would strengthen the paper, but our compute resources are modest. We focused on the ablation studies that are maximally informative about our new model, such as verifying which of the projections are necessary to use (see Tab. 6 in the Appendix).
>
> > The paper mentions "Our preliminary experiments" several times (e.g., lines 91, 473) without providing citations or section indicators. Could you clarify these references to improve the paper's clarity and reproducibility?
>
> We refer to “preliminary experiments” as experiments not included in the paper, but done before the final experimental protocol is decided (to the best of our knowledge, this is a rather common terminology). They are typically conducted in order to determine the final model/protocol to be studied and presented in the paper, i.e., the final setting on which we spend our compute resources. These are not necessarily done in the exact same setting as the experiments reported in the paper and, thus, are not directly comparable. As we mention in L512-513 of the appendix, we have done an order of magnitude more of these experiments than the ones included in the paper.
>
> We hope our response above brings clarifications to all the concerns that the reviewer has raised. Thank you very much.

---

> > ### Comment · Reviewer_HTZS · 2024-08-12
> >
> > The review committee extends its sincere appreciation to the authors for their comprehensive and insightful responses to the critiques provided. The diligence demonstrated in addressing the concerns raised and elucidating various aspects of the research is highly commendable. In light of these thorough explanations, the committee is inclined to view the manuscript more favorably.
> > While substantial progress has been made in addressing the initial reservations, there remains one salient point that warrants further deliberation:
> >
> > Regarding the quantitative analysis of attention maps:
> >
> > - I appreciate the authors' insight that it's challenging to draw definitive conclusions from raw entropy measurements of attention maps. The perspective that downstream task performance is ultimately the most meaningful metric for assessing the adequacy of attention maps is well-taken. However, the paper's current statements about attention maps being "qualitatively similar" (Lines 227, 236) could potentially be expanded upon to provide more valuable insights. Perhaps the authors could consider alternative methods or metrics that could provide more meaningful insights into the relationship between attention map characteristics and model performance.
> > - There seems to be an underlying assumption that similar attention maps between the baseline model and the MoE model could explain why the MoE model performs well. If this is indeed a key point, it might be beneficial to explore this assumption further. For instance, is there a way to investigate any potential correlation between the similarity of attention maps and downstream task performance?
> >
> > By addressing these points, the paper could potentially offer deeper insights into the workings of the SwitchHead model and strengthen its contributions to the field.

---

> > > ### Author Response · Authors · 2024-08-12
> > >
> > > Thank you very much for your response! We are glad to hear that the reviewer found our response useful!
> > >
> > > Regarding the first point:
> > >
> > > > Perhaps the authors could consider alternative methods or metrics that could provide more meaningful insights into the relationship between attention map characteristics and model performance
> > >
> > > We agree with the reviewer that going beyond this qualitative findings would be valuable. However, currently, we do not have any convincingly good metrics for this purpose.
> > >
> > > Regarding the second point:
> > >
> > > > There seems to be an underlying assumption that similar attention maps between the baseline model and the MoE model could explain why the MoE model performs well. If this is indeed a key point, …
> > >
> > > We think there is a misunderstanding here. We did not intend to make such an assumption. The sole goal of our attention map analysis is to qualitatively examine how the reduced number of heads (a characteristic of Switchhead) affects the global attention patterns. Here for ListOps, we observe that a high-level pattern is similar to the baseline. This is a purely behavioral comparison between our model and the baseline; we do not use this observation to justify the performance of our model. We will clarify this in the future version of the paper.
> > >
> > > We hope this response provides further clarifications about the purpose of our attention map analysis.

---

### Official Review · Reviewer_58yB · 2024-07-13

**Soundness:** 4
**Presentation:** 3
**Contribution:** 3
**Rating:** 7
**Confidence:** 3

**Summary:**

This work proposes a novel Multi-Head-Attention (MHA) mechanism which is more efficient than the standard MHA used in most transformers. The method---called SwitchHead---is relying on Mixture-of-Experts (MoEs) to save computation while retaining the same model capacity and performance. To achieve this, instead of naively selecting which head should be computed for each token, a small number of heads are **always** computed, and MoEs are used to modulate the value and output projections for each head. Using multiple transformer variants and datasets, they empirically show how SwichHead transformers reach a similar perplexity as the baseline, for the same number of parameters, while requiring fewer operations and memory. They also show how their approach can yield a significant speedup during training.

**Strengths:**

The paper is well written and well motivated.
While the components of the proposed method are not particularly novel, they are combined in a simple yet novel way.
I find the experiments convincing. Matching the parameters and perplexity allows to clearly appreciate the gains in memory and MACs. The results of the MAC-matched experiments and the time comparisons further reinforce the potential impact of the proposed method.

**Weaknesses:**

- As mentioned in the limitations section, the models are relatively small.
- I could not find the sequence length used in your experiments, which prevents me from computing the number of tokens used during training.
- I find it odd that dropout is used for the baseline but not for the switchAll models, on C4 I'm expecting dropout to slow down learning.
- What is explaining the differences in the transformers' PPLs in table 4 compared to the PPLs in table 2?

**Questions:**

See above.

**Limitations:**

Limitations have been discussed adequately.

---

> ### Author Rebuttal · Authors · 2024-08-06
>
> We would like to thank the reviewer for their insightful review and for positive comments on the clarity and methodology of the paper. Please find our responses as follows:
>
> > I could not find the sequence length used in your experiments, which prevents me from computing the number of tokens used during training.
>
> The sequence length is T in Tab. 6 in the appendix.
>
> > I find it odd that dropout is used for the baseline but not for the switchAll models, on C4 I'm expecting dropout to slow down learning.
>
> The dropout is used in the switchAll models but only in the feedforward blocks. We do not use dropout in the attention components. In some sense, a dropout-like behavior is naturally provided by the expert selection mechanism: in the early stages of the training, the expert selection is “random” (not trained yet), and in all stages, only a small percentage of the total number of experts is selected (the others can be considered as "dropped out"). Moreover, the baseline only uses dropout on the projections, and neither model uses dropout on the attention scores.
>
> > What is explaining the differences in the transformers' PPLs in table 4 compared to the PPLs in table 2?
>
> They correspond to runs with different seeds, for historical reasons. We will update Tab. 2 in the next version to avoid this confusion.
>
> We believe our response above resolves the concerns that the reviewer has raised. Thank you very much.

---

> > ### Comment · Reviewer_58yB · 2024-08-11
> >
> > I thank the authors for these clarifications.

---

### Decision · Program_Chairs · 2024-09-25

**Decision:**

Accept (poster)

**Comment:**

SwitchHead demonstrates impressive performance improvements in language modeling tasks, showcasing the potential of MoE in self-attention layers. Despite some minor limitations, the reviewers generally agree on the technical soundness and significance of the work.
Given the overall positive reviews, I recommend accepting this paper. Addressing the weaknesses identified during the review process, such as expanding the evaluation to additional tasks and providing more quantitative analysis of attention maps, would further enhance the paper's contribution.